# Structure-Blind Signal Recovery

**Dmitry Ostrovsky**[*]  **Zaid Harchaoui**[†]  **Anatoli Juditsky**[*]  **Arkadi Nemirovski**[‡]
`firstname.lastname@imag.fr`

## Abstract

We consider the problem of recovering a signal observed in Gaussian noise. If the set of signals is convex and compact, and can be specified beforehand, one can use classical linear estimators that achieve a risk within a constant factor of the minimax risk. However, when the set is unspecified, designing an estimator that is blind to the hidden structure of the signal remains a challenging problem. We propose a new family of estimators to recover signals observed in Gaussian noise. Instead of specifying the set where the signal lives, we assume the existence of a well-performing linear estimator. Proposed estimators enjoy exact oracle inequalities and can be efficiently computed through convex optimization. We present several numerical illustrations that show the potential of the approach.

## 1 Introduction

We consider the problem of recovering a *complex-valued signal* $(x_t)_{t \in \mathbb{Z}}$ from the noisy observations

$$y_\tau = x_\tau + \sigma \zeta_\tau, \quad -n \leq \tau \leq n. \tag{1}$$

Here $n \in \mathbb{Z}_+$, and $\zeta_\tau \sim \mathbb{CN}(0,1)$ are i.i.d. standard complex-valued Gaussian random variables, meaning that $\zeta_0 = \xi_0^1 + \imath \xi_0^2$ with i.i.d. $\xi_0^1, \xi_0^2 \sim \mathcal{N}(0,1)$. Our goal is to recover $x_t$, $0 \leq t \leq n$, given the sequence of observations $y_{t-n}, ..., y_t$ up to instant $t$, a task usually referred to as (pointwise) *filtering* in machine learning, statistics, and signal processing [5].

The traditional approach to this problem considers *linear estimators*, or linear filters, which write as

$$\widehat{x}_t = \sum_{\tau=0}^{n} \phi_\tau y_{t-\tau}, \quad 0 \leq t \leq n.$$

Linear estimators have been thoroughly studied in various forms, they are both theoretically attractive [7, 3, 2, 16, 17, 11, 13] and easy to use in practice. If the set $\mathcal{X}$ of signals is well-specified, one can usually compute a (nearly) minimax on $\mathcal{X}$ linear estimator in a closed form. In particular, if $\mathcal{X}$ is a class of smooth signals, such as a Hölder or a Sobolev ball, then the corresponding estimator is given by the kernel estimator with the properly set bandwidth parameter [16] and is minimax among all possible estimators. Moreover, as shown by [6, 2], if only $\mathcal{X}$ is convex, compact, and centrally symmetric, the risk of the best linear estimator of $x_t$ is within a small constant factor of the minimax risk over $\mathcal{X}$. Besides, if the set $\mathcal{X}$ can be specified in a computationally tractable way, which clearly is still a weaker assumption than classical smoothness assumptions, the best linear estimator can be efficiently computed by solving a convex optimization problem on $\mathcal{X}$. In other words, given a computationally tractable set $\mathcal{X}$ on the input, one can *compute* a nearly-minimax linear estimator and the corresponding (nearly-minimax) risk over $\mathcal{X}$. The strength of this approach, however, comes at

---

[*]LJK, University of Grenoble Alpes, 700 Avenue Centrale, 38401 Domaine Universitaire de Saint-Martin-d'Hères, France.

[†]University of Washington, Seattle, WA 98195, USA.

[‡]Georgia Institute of Technology, Atlanta, GA 30332, USA.

a price: the set $\mathcal{X}$ still must be specified beforehand. Therefore, when one faces a recovery problem *without any prior knowledge of $\mathcal{X}$*, this approach cannot be implemented.

We adopt here a novel approach to filtering, which we refer to as *structure-blind recovery*. While we do not require $\mathcal{X}$ to be specified beforehand, we assume that there exists a *linear oracle* – a well-performing linear estimator of $x_t$. Previous works [8, 10, 4], following a similar philosophy, proved that one can efficiently adapt to the linear oracle filter of length $m = O(n)$ if the corresponding filter $\phi$ is *time-invariant*, i.e. it recovers the target signal uniformly well in the $O(n)$-sized neighbourhood of $t$, and if its $\ell_2$-norm is small – bounded by $\rho/\sqrt{m}$ for a moderate $\rho \geq 1$. The adaptive estimator is computed by minimizing the $\ell_\infty$-norm of the filter discrepancy, in the Fourier domain, under the constraint on the $\ell_1$-norm of the filter in the Fourier domain. Put in contrast to the oracle linear filter, the price for adaptation is proved to be $O(\rho^3\sqrt{\ln n})$, with the lower bound of $O(\rho\sqrt{\ln n})$ [8, 4].

We make the following contributions:

- we propose a new family of recovery methods, obtained by solving a least-squares problem constrained or penalized by the $\ell_1$-norm of the filter in the Fourier domain;

- we prove exact oracle inequalities for the $\ell_2$-risk of these methods;

- we show that the price for adaptation improves upon previous works [8, 4] to $O(\rho^2\sqrt{\ln n})$ for the point-wise risk and to $O(\rho\sqrt{\ln n})$ for the $\ell_2$-risk.

- we present numerical experiments that show the potential of the approach on synthetic and real-world images and signals.

Before presenting the theoretical results, let us introduce the notation we use throughout the paper.

**Filters**    Let $\mathbb{C}(\mathbb{Z})$ be the linear space of all two-sided complex-valued sequences $x = \{x_t \in \mathbb{C}\}_{t\in\mathbb{Z}}$. For $k, k' \in \mathbb{Z}$ we consider finite-dimensional subspaces

$$\mathbb{C}(\mathbb{Z}_k^{k'}) = \{x \in \mathbb{C}(\mathbb{Z}) : \quad x_t = 0, \quad t \notin [k, k']\}.$$

It is convenient to identify $m$-dimensional complex vectors, $m = k' - k + 1$, with elements of $\mathbb{C}(\mathbb{Z}_k^{k'})$ by means of the notation:

$$x_k^{k'} := [x_k; ...; x_{k'}] \in \mathbb{C}^{k'-k+1}.$$

We associate to linear mappings $\mathbb{C}(\mathbb{Z}_k^{k'}) \to \mathbb{C}(\mathbb{Z}_j^{j'})$ $(j'-j+1) \times (k'-k+1)$ matrices with complex entries. The *convolution* $u * v$ of two sequences $u, v \in \mathbb{C}(\mathbb{Z})$ is a sequence with elements

$$[u * v]_t = \sum_{\tau\in\mathbb{Z}} u_\tau v_{t-\tau}, \quad t \in \mathbb{Z}.$$

Given observations (1) and $\varphi \in \mathbb{C}(\mathbb{Z}_0^m)$ consider the *(left) linear estimation* of $x$ associated with *filter $\varphi$*:

$$\widehat{x}_t = [\varphi * y]_t$$

($\widehat{x}_t$ is merely a kernel estimate of $x_t$ by a kernel $\varphi$ supported on $[0, ..., m]$).

**Discrete Fourier transform**    We define the unitary *Discrete Fourier transform* (DFT) operator $F_n : \mathbb{C}^{n+1} \to \mathbb{C}^{n+1}$ by

$$z \mapsto F_n z, \quad [F_n z]_k = (n+1)^{-1/2} \sum_{t=0}^{n} z_t\, e^{\frac{2\pi\imath kt}{n+1}}, \quad 0 \leq k \leq n.$$

The *inverse Discrete Fourier transform* (iDFT) operator $F_n^{-1}$ is given by $F_n^{-1} := F_n^H$ (here $A^H$ stands for Hermitian adjoint of $A$). By the Fourier inversion theorem, $F_n^{-1}(F_n z) = z$.

We denote $\|\cdot\|_p$ usual $\ell_p$-norms on $\mathbb{C}(\mathbb{Z})$: $\|x\|_p = (\sum_{t\in\mathbb{Z}} |x_t|^p)^{1/p}$, $p \in [1, \infty]$. Usually, the argument will be finite-dimensional – an element of $\mathbb{C}(\mathbb{Z}_k^{k'})$; we reserve the special notation

$$\|x\|_{n,p} := \|x_0^n\|_p.$$

Furthermore, DFT allows to equip $\mathbb{C}(\mathbb{Z}_0^n)$ with the norms associated with $\ell_p$-norms in the spectral domain:

$$\|x\|_{n,p}^* := \|x_0^n\|_p^* := \|F_n x_0^n\|_p, \quad p \in [1, \infty];$$

note that unitarity of the DFT implies the Parseval identity: $\|x\|_{n,2} = \|x\|_{n,2}^*$.

Finally, $c$, $C$, and $C'$ stand for generic absolute constants.

## 2 Oracle inequality for constrained recovery

Given observations (1) and $\overline{\varrho} > 0$, we first consider the *constrained recovery* $\widehat{x}_{\mathrm{con}}$ given by

$$[\widehat{x}_{\mathrm{con}}]_t = [\widehat{\varphi} * y]_t, \quad t = 0, ..., n,$$

where $\widehat{\varphi}$ is an optimal solution of the constrained optimization problem

$$\min_{\varphi \in \mathbb{C}(\mathbb{Z}_0^n)} \left\{ \|y - \varphi * y\|_{n,2} : \|\varphi\|_{n,1}^* \leq \overline{\varrho}/\sqrt{n+1} \right\}. \tag{2}$$

The constrained recovery estimator minimizes a least-squares fit criterion under a constraint on $\|\varphi\|_{n,1}^* = \|F_n \varphi_0^n\|_1$, that is an $\ell_1$ constraint on the discrete Fourier transform of the filter. While the least-squares objective naturally follows from the Gaussian noise assumption, the constraint can be motivated as follows.

**Small-error linear filters** Linear filter $\varphi^o$ with a small $\ell_1$ norm in the spectral domain and small recovery error exists, essentially, whenever there exists a linear filter with small recovery error [8, 4]. Indeed, let us say that $x \in \mathbb{C}(\mathbb{Z}_0^n)$ is *simple* [4] *with parameters* $m \in \mathbb{Z}_+$ *and* $\rho \geq 1$ if there exists $\phi^o \in \mathbb{C}(\mathbb{Z}_0^m)$ such that for all $-m \leq \tau \leq 2m$,

$$\left[ \mathbf{E} \left\{ |x_\tau - [\phi^o * y]_\tau|^2 \right\} \right]^{1/2} \leq \frac{\sigma \rho}{\sqrt{m+1}}. \tag{3}$$

In other words, $x$ is $(m, \rho)$-simple if there exists a hypothetical filter $\phi^o$ of the length at most $m+1$ which recovers $x_\tau$ with squared risk uniformly bounded by $\frac{\sigma^2 \rho^2}{m+1}$ in the interval $-m \leq \tau \leq 2m$. Note that (3) clearly implies that $\|\phi^o\|_2 \leq \rho/\sqrt{m+1}$, and that $|[x - \phi^o * x]_\tau| \leq \sigma\rho/\sqrt{m+1}$ $\forall \tau$, $-m \leq \tau \leq 2m$. Now, let $n = 2m$, and let

$$\varphi^o = \phi^o * \phi^o \in \mathbb{C}^{n+1}.$$

As proved in [15, Appendix C], we have

$$\|\varphi^o\|_{n,1}^* \leq 2\rho^2/\sqrt{n+1}, \tag{4}$$

and, for a moderate absolute constant $c$,

$$\|x - \varphi^o * y\|_{n,2} \leq c\sigma\rho^2 \sqrt{1 + \ln[1/\alpha]} \tag{5}$$

with probability $1 - \alpha$. To summarize, *if $x$ is $(m, \rho)$-simple, i.e., when there exists a filter $\phi^o$ of length $\leq m+1$ which recovers $x$ with small risk on the interval $[-m, 2m]$, then the filter $\varphi^o = \phi^o * \phi^o$ of the length at most $n+1$, with $n = 2m$, has small norm $\|\varphi^o\|_{n,1}^*$ and recovers the signal $x$ with (essentially the same) small risk on the interval $[0, n]$.*

**Hidden structure** The constrained recovery estimator is completely blind to a possible hidden structure of the signal, yet can seamlessly adapt to it when such a structure exists, in a way that we can rigorously establish. Using the right-shift operator on $\mathbb{C}(\mathbb{Z})$, $[\Delta x]_t = x_{t-1}$, we formalize the hidden structure as an unknown shift-invariant linear subspace of $\mathbb{C}(\mathbb{Z})$, $\Delta \mathcal{S} = \mathcal{S}$, of a small dimension $s$. We do not assume that $x$ belongs to that subspace. Instead, we make a more general assumption that $x$ is *close* to this subspace, that is, it may be decomposed into a sum of a component that lies in the subspace and a component whose norm we can control.

**Assumption A** *We suppose that $x$ admits the decomposition*

$$x = x^{\mathcal{S}} + \varepsilon, \quad x^{\mathcal{S}} \in \mathcal{S},$$

*where $\mathcal{S}$ is an (unknown) shift-invariant, $\Delta \mathcal{S} = \mathcal{S}$, subspace of $\mathbb{C}(\mathbb{Z})$ of dimension $s$, $1 \le s \le n+1$, and $\varepsilon$ is "small", namely,*

$$\|\Delta^\tau \varepsilon\|_{n,2} \le \sigma \varkappa, \quad 0 \le \tau \le n.$$

Shift-invariant subspaces of $\mathbb{C}(\mathbb{Z})$ are exactly the sets of solutions of homogeneous linear difference equations with polynomial operators. This is summarized by the following lemma (we believe it is a known fact; for completeness we provide a proof in [15, Appendix C]).

**Lemma 2.1.** *Solution set of a homogeneous difference equation with a polynomial operator $p(\Delta)$,*

$$[p(\Delta)x]_t = \left[ \sum_{\tau=0}^{s} p_\tau x_{t-\tau} \right] = 0, \quad t \in \mathbb{Z}, \tag{6}$$

*with $\deg(p(\cdot)) = s$, $p(0) = 1$, is a shift-invariant subspace of $\mathbb{C}(\mathbb{Z})$ of dimension $s$. Conversely, any shift-invariant subspace $\mathcal{S} \subset \mathbb{C}(\mathbb{Z})$, $\Delta \mathcal{S} \subseteq \mathcal{S}$, $\dim(\mathcal{S}) = s < \infty$, is the set of solutions of some homogeneous difference equation (6) with $\deg(p(\cdot)) = s$, $p(0) = 1$. Moreover, such $p(\cdot)$ is unique.*

On the other hand, for any polynomial $p(\cdot)$, solutions of (6) are exponential polynomials [**?**] with frequencies determined by the roots of $p(\cdot)$. For instance, discrete-time polynomials $x_t = \sum_{k=0}^{s-1} c_k t^k$, $t \in \mathbb{Z}$ of degree $s-1$ (that is, exponential polynomials with all zero frequencies) form a linear space of dimension $s$ of solutions of the equation (6) with a polynomial $p(\Delta) = (1 - \Delta)^s$ with a unique root of multiplicity $s$, having coefficients $p_k = (-1)^k \binom{s}{k}$. Naturally, signals which are close, in the $\ell_2$ distance, to discrete-time polynomials are Sobolev-smooth functions sampled over the regular grid [10]. Sum of harmonic oscillations $x_t = \sum_{k=1}^{s} c_k e^{\imath \omega_k t}$, $\omega_k \in [0, 2\pi)$ being all different, is another example; here, $p(\Delta) = \prod_{k=1}^{s} (1 - e^{\imath \omega_k} \Delta)$.

We can now state an oracle inequality for the constrained recovery estimator; see [15, Appendix B].

**Theorem 2.1.** *Let $\bar{\varrho} \ge 1$, and let $\varphi^o \in \mathbb{C}(\mathbb{Z}_0^n)$ be such that*

$$\|\varphi^o\|_{n,1}^* \le \bar{\varrho}/\sqrt{n+1}.$$

*Suppose that Assumption A holds for some $s \in \mathbb{Z}_+$ and $\varkappa < \infty$. Then for any $\alpha$, $0 < \alpha \le 1$, it holds with probability at least $1 - \alpha$:*

$$\|x - \widehat{x}_{\mathrm{con}}\|_{n,2} \le \|x - \varphi^o * y\|_{n,2} + C\sigma \sqrt{s + \bar{\varrho}\left(\varkappa\sqrt{\ln[1/\alpha]} + \ln[n/\alpha]\right)}. \tag{7}$$

When considering *simple* signals, Theorem 2.1 gives the following.

**Corollary 2.1.** *Assume that signal $x$ is $(m, \rho)$-simple, $\rho \ge 1$ and $m \in \mathbb{Z}_+$. Let $n = 2m$, $\bar{\varrho} \ge 2\rho^2$, and let Assumption A hold for some $s \in \mathbb{Z}_+$ and $\varkappa < \infty$. Then for any $\alpha$, $0 < \alpha \le 1$, it holds with probability at least $1 - \alpha$:*

$$\|x - \widehat{x}_{\mathrm{con}}\|_{n,2} \le C\sigma\rho^2\sqrt{\ln[1/\alpha]} + C'\sigma\sqrt{s + \bar{\varrho}\left(\varkappa\sqrt{\ln[1/\alpha]} + \ln[n/\alpha]\right)}.$$

**Adaptation and price** The price for adaptation in Theorem 2.1 and Corollary 2.1 is determined by three parameters: the bound on the filter norm $\bar{\varrho}$, the deterministic error $\varkappa$, and the subspace dimension $s$. Assuming that the signal to recover is simple, and that $\bar{\varrho} = 2\rho^2$, let us compare the magnitude of the oracle error to the term of the risk which reflects "price of adaptation". Typically (in fact, in all known to us cases of recovery of signals from a shift-invariant subspace), the parameter $\rho$ is at least $\sqrt{s}$. Therefore, the bound (5) implies the "typical bound" $O(\sigma\sqrt{\gamma}\rho^2) = \sigma s\sqrt{\gamma}$ for the term $\|x - \varphi^o * y\|_{n,2}$ (we denote $\gamma = \ln(1/\alpha)$). As a result, for instance, in the "parametric situation", when the signal belongs or is very close to the subspace, that is when $\varkappa = O(\ln(n))$, the price of adaptation $O\left(\sigma[s + \rho^2(\gamma + \sqrt{\gamma}\ln n)]^{1/2}\right)$ is much smaller than the bound on the oracle error. In the "nonparametric situation", when $\varkappa = O(\rho^2)$, the price of adaptation has the same order of magnitude as the oracle error.

Finally, note that under the premise of Corollary 2.1 we can also bound the pointwise error. We state the result for $\bar{\varrho} = 2\rho^2$ for simplicity; the proof can be found in [15, Appendix B].

**Theorem 2.2.** *Assume that signal $x$ is $(m, \rho)$-simple, $\rho \geq 1$ and $m \in \mathbb{Z}_+$. Let $n = 2m$, $\overline{\varrho} = 2\rho^2$, and let Assumption A hold for some $s \in \mathbb{Z}_+$ and $\varkappa < \infty$. Then for any $\alpha, 0 < \alpha \leq 1$, the constrained recovery $\widehat{x}_{\mathrm{con}}$ satisfies*

$$|x_n - [\widehat{x}_{\mathrm{con}}]_n| \leq C \frac{\sigma\rho}{\sqrt{m+1}} \left[ \rho^2 \sqrt{\ln[n/\alpha]} + \rho\sqrt{\varkappa\sqrt{\ln[1/\alpha]}} + \sqrt{s} \right].$$

## 3  Oracle inequality for penalized recovery

To use the constrained recovery estimator with a provable guarantee, see e.g. Theorem 2.1, one must know the norm of a small-error linear filter $\varrho$, or at least have an upper bound on it. However, if this parameter is unknown, but instead the noise variance is known (or can be estimated from data), we can build a more practical estimator that still enjoys an oracle inequality.

The *penalized recovery* estimator $[\widehat{x}_{\mathrm{pen}}]_t = [\widehat{\varphi} * y]_t$ is an optimal solution to a regularized least-squares minimization problem, where the regularization penalizes the $\ell_1$-norm of the filter in the Fourier domain:

$$\widehat{\varphi} \in \underset{\varphi \in \mathbb{C}(\mathbb{Z}_0^n)}{\mathrm{Argmin}} \left\{ \|y - \varphi * y\|_{n,2}^2 + \lambda\sqrt{n+1} \, \|\varphi\|_{n,1}^* \right\}. \tag{8}$$

Similarly to Theorem 2.1, we establish an oracle inequality for the penalized recovery estimator.

**Theorem 3.1.** *Let Assumption A hold for some $s \in \mathbb{Z}_+$ and $\varkappa < \infty$, and let $\varphi^o \in \mathbb{C}(\mathbb{Z}_0^n)$ satisfy $\|\varphi^o\|_{n,1}^* \leq \varrho/\sqrt{n+1}$ for some $\varrho \geq 1$.*

$1^o$. *Suppose that the regularization parameter of penalized recovery $\widehat{x}_{\mathrm{pen}}$ satisfies $\lambda \geq \underline{\lambda}$,*

$$\underline{\lambda} := 60\sigma^2 \ln[63n/\alpha].$$

*Then, for $0 < \alpha \leq 1$, it holds with probability at least $1 - \alpha$:*

$$\|x - \widehat{x}_{\mathrm{pen}}\|_{n,2} \leq \|x - \varphi^o * y\|_{n,2} + C\sqrt{\varrho\lambda} + C'\sigma\sqrt{s + (\widehat{\varrho}+1)\varkappa\sqrt{\ln[1/\alpha]}},$$

*where $\widehat{\varrho} := \sqrt{n+1} \, \|\widehat{\varphi}\|_{n,1}^*$.*

$2^o$. *Moreover, if $\varkappa \leq \bar{\varkappa}$,*

$$\bar{\varkappa} := \frac{10 \ln[42n/\alpha]}{\sqrt{\ln[16/\alpha]}},$$

*and $\lambda \geq 2\underline{\lambda}$, one has*

$$\|x - \widehat{x}_{\mathrm{pen}}\|_{n,2} \leq \|x - \varphi^o * y\|_{n,2} + C\sqrt{\varrho\lambda} + C'\sigma\sqrt{s}.$$

The proof closely follows that of Theorem 2.1 and can also be found in [15, Appendix B].

## 4  Discussion

There is some redundancy between "simplicity" of a signal, as defined by (3), and Assumption A. Usually a simple signal or image $x$ is also close to a low-dimensional subspace of $\mathbb{C}(\mathbb{Z})$ (see, e.g., [10, section 4]), so that Assumption A holds "automatically". Likewise, $x$ is "almost" simple when it is close to a low-dimensional time-invariant subspace. Indeed, if $x \in \mathbb{C}(\mathbb{Z})$ *belongs to* $\mathcal{S}$, i.e. Assumption A holds with $\varkappa = 0$, one can easily verify that for $n \geq s$ there exists a filter $\phi^o \in \mathbb{C}(\mathbb{Z}_{-n}^n)$ such that

$$\|\phi^o\|_2 \leq \sqrt{s/(n+1)}, \text{ and } x_\tau = [\phi^o * x]_\tau, \ \tau \in \mathbb{Z}. \tag{9}$$

See [15, Appendix C] for the proof. This implies that $x$ can be recovered efficiently from observations (1):

$$\left[ \mathbf{E}\{|x_\tau - [\phi^o * y]_\tau|^2\} \right]^{1/2} \leq \sigma\sqrt{\frac{s}{n+1}}.$$

In other words, if instead of the filtering problem we were interested in the *interpolation* problem of recovering $x_t$ given $2n+1$ observations $y_{t-n}, ..., y_{t+n}$ on the left *and* on the right of $t$, Assumption

A would imply a kind of simplicity of $x$. On the other hand, it is clear that Assumption A is not sufficient to imply the simplicity of $x$ "with respect to the filtering", in the sense of the definition we use in this paper, when we are allowed to use only observations on the left of $t$ to compute the estimation of $x_t$. Indeed, one can see, for instance, that already signals from the parametric family $\mathcal{X}_\alpha = \{x \in \mathbb{C}(\mathbb{Z}) : x_\tau = c\alpha^\tau, c \in \mathbb{C}\}$, with a given $|\alpha| > 1$, which form a one-dimensional space of solutions of the equation $x_\tau = \alpha x_{\tau-1}$, cannot be estimated with small risk at $t$ using only observations on the left of $t$ (unless $c = 0$), and thus are not simple in the sense of (3).

Of course, in the above example, the "difficulty" of the family $\mathcal{X}_\alpha$ is due to instability of solutions of the difference equation which explode when $\tau \to +\infty$. Note that signals $x \in \mathcal{X}_\alpha$ with $|\alpha| \leq 1$ (linear functions, oscillations, or damped oscillations) are simple. More generally, suppose that $x$ satisfies a difference equation of degree $s$:

$$0 = p(\Delta)x_\tau \left[ = \sum_{i=0}^{s} p_i x_{\tau-i} \right], \tag{10}$$

where $p(z) = \sum_{i=0}^{s} p_i z^i$ is the corresponding characteristic polynomial and $\Delta$ is the right shift operator. When $p(z)$ is unstable – has roots *inside* the unit circle – (depending on "initial conditions") the set of solutions to the equation (10) contains difficult to filter signals. Observe that stability of solutions is related to the direction of the time axis; when the characteristic polynomial $p(z)$ has roots *outside* the unit circle, the corresponding solutions may be "left unstable" – increase exponentially when $\tau \to -\infty$. In this case "right filtering" – estimating $x_\tau$ using observations on the right of $\tau$ – will be difficult. A special situation where interpolation and filtering is always simple arises when the characteristic polynomial of the difference equation has all its roots on the unit circle. In this case, solutions to (10) are "generalized harmonic oscillations" (harmonic oscillations modulated by polynomials), and such signals are known to be simple. Theorem 4.1 summarizes the properties of the solutions of (10) in this particular case; see [15, Appendix C] for the proof.

**Theorem 4.1.** *Let $s$ be a positive integer, and let $p = [p_0; ...; p_s] \in \mathbb{C}^{s+1}$ be such that the polynomial $p(z) = \sum_{i=0}^{s} p_i z^i$ has all its roots on the unit circle. Then for every integer $m$ satisfying*

$$m \geq m(s) := Cs^2 \ln(s+1),$$

*one can point out $q \in \mathbb{C}^{m+1}$ such that any solution to* (10) *satisfies*

$$x_\tau = [q * x]_\tau, \quad \forall \tau \in \mathbb{Z},$$

*and*

$$\|q\|_2 \leq \rho(s,m)/\sqrt{m} \quad where \quad \rho(s,m) = C' \min \left\{ s^{3/2}\sqrt{\ln s}, \ s\sqrt{\ln[ms]} \right\}. \tag{11}$$

## 5   Numerical experiments

We present preliminary results on simulated data of the proposed adaptive signal recovery methods in several application scenarios. We compare the performance of the *penalized $\ell_2$-recovery* of Sec. 3 to that of the Lasso recovery of [1] in signal and image denoising problems. Implementation details for the penalized $\ell_2$-recovery are given in Sec. 6. Discussion of the discretization approach underlying the competing Lasso method can be found in [1, Sec. 3.6].

We follow the same methodology in both signal and image denoising experiments. For each level of the signal-to-noise ratio SNR $\in \{1, 2, 4, 8, 16\}$, we perform $N$ Monte-Carlo trials. In each trial, we generate a random signal $x$ on a regular grid with $n$ points, corrupted by the i.i.d. Gaussian noise of variance $\sigma^2$. The signal is normalized: $\|x\|_2 = 1$ so $\text{SNR}^{-1} = \sigma\sqrt{n}$. We set the regularization penalty in each method as follows. For penalized $\ell_2$-recovery (8), we use $\lambda = 2\sigma^2 \log[63n/\alpha]$ with $\alpha = 0.1$. For Lasso [1], we use the common setting $\lambda = \sigma\sqrt{2\log n}$. We report experimental results by plotting the $\ell_2$-error $\|\widehat{x} - x\|_2$, averaged over $N$ Monte-Carlo trials, versus the inverse of the signal-to-noise ratio $\text{SNR}^{-1}$.

**Signal denoising**   We consider denoising of a one-dimensional signal in two different scenarios, fixing $N = 100$ and $n = 100$. In the *RandomSpikes* scenario, the signal is a sum of 4 harmonic oscillations, each characterized by a spike of a random amplitude at a random position in the continuous frequency domain $[0, 2\pi]$. In the *CoherentSpikes* scenario, the same number of spikes is

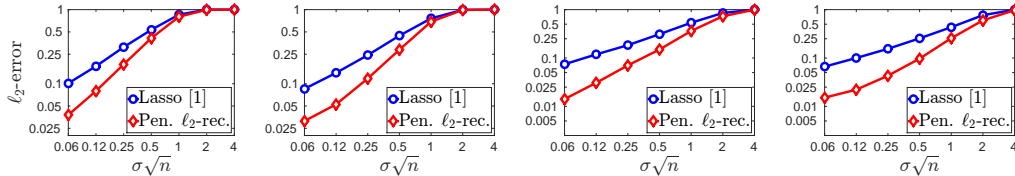

Figure 1: Signal and image denoising in different scenarios, left to right: *RandomSpikes*, *CoherentSpikes*, *RandomSpikes-2D*, and *CoherentSpikes-2D*. The steep parts of the curves on high noise levels correspond to observations being thresholded to zero.

sampled by pairs. Spikes in each pair have the same amplitude and are separated by only $0.1$ of the DFT bin $2\pi/n$ which could make recovery harder due to high signal coherency. However, in practice we found *RandomSpikes* to be slightly harder than *CoherentSpikes* for both methods, see Fig. 1. As Fig. 1 shows, the proposed penalized $\ell_2$-recovery outperforms the Lasso method for all noise levels. The performance gain is particularly significant for high signal-to-noise ratios.

**Image Denoising**   We now consider recovery of an unknown regression function $f$ on the regular grid on $[0, 1]^2$ given the noisy observations:

$$y_\tau = x_\tau + \sigma\zeta_\tau, \quad \tau \in \{0, 1, ..., m - 1\}^2, \tag{12}$$

where $x_\tau = f(\tau/m)$. We fix $N = 40$, and the grid dimension $m = 40$; the number of samples is then $n = m^2$. For the penalized $\ell_2$-recovery, we implement the blockwise denoising strategy (see Appendix for the implementation details) with just one block for the entire image. We present additional numerical illustrations in the supplementary material.

We study three different scenarios for generating the ground-truth signal in this experiment. The first two scenarios, *RandomSpikes-2D* and *CoherentSpikes-2D*, are two-dimensional counterparts of those studied in the signal denoising experiment: the ground-truth signal is a sum of $4$ harmonic oscillations in $\mathbb{R}^2$ with random frequencies and amplitudes. The separation in the *CoherentSpikes-2D* scenario is $0.2\pi/m$ in each dimension of the torus $[0, 2\pi]^2$. The results for these scenarios are shown in Fig. 1. Again, the proposed penalized $\ell_2$-recovery outperforms the Lasso method for all noise levels, especially for high signal-to-noise ratios.

In scenario *DimensionReduction-2D* we investigate the problem of estimating a function with a hidden low-dimensional structure. We consider the single-index model of the regression function:

$$f(t) = g(\theta^T t), \quad g(\cdot) \in \mathcal{S}_\beta^1(1). \tag{13}$$

Here, $\mathcal{S}_\beta^1(1) = \{g : \mathbb{R} \to \mathbb{R}, \|g^{(\beta)}(\cdot)\|_2 \leq 1\}$ is the Sobolev ball of smooth periodic functions on $[0, 1]$, and the unknown structure is formalized as the direction $\theta$. In our experiments we sample the direction $\theta$ uniformly at random and consider different values of the smoothness index $\beta$. If it is known a priori that the regression function possesses the structure (13), and only the index is unknown, one can use estimators attaining "one-dimensional" rates of recovery; see e.g. [12] and references therein. In contrast, our recovery algorithms are not aware of the underlying structure but might still adapt to it.

As shown in Fig. 2, the $\ell_2$-recovery performs well in this scenario despite the fact that the available theoretical bounds are pessimistic. For example, the signal (13) with a smooth $g$ can be approximated by a small number of harmonic oscillations in $\mathbb{R}^2$. As follows from the proof of [9, Proposition 10] combined with Theorem 4.1, for a sum of $k$ harmonic oscillations in $\mathbb{R}^d$ one can point out a reproducing linear filter with $\varrho(k) = O(k^{2d})$ (neglecting the logarithmic factors), i.e. the theoretical guarantee is quite conservative for small values of $\beta$.

## 6   Details of algorithm implementation

Here we give a brief account of some techniques and implementation tricks exploited in our codes.

**Solving the optimization problems**   Note that the optimization problems (2) and (8) underlying the proposed recovery algorithms are well structured Second-Order Conic Programs (SOCP) and

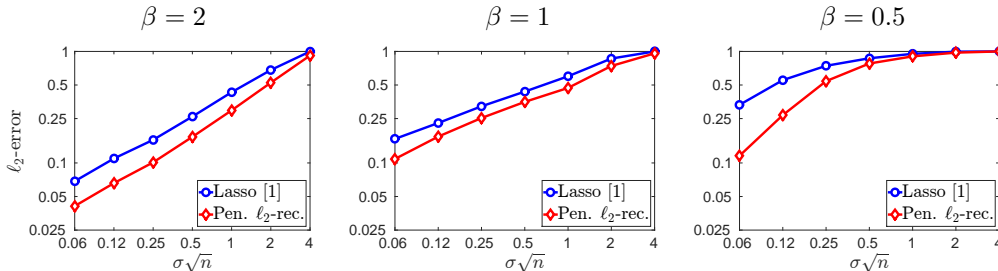

Figure 2: Image denoising in *DimensionReduction* scenario; smoothness decreases from left to right.

can be solved using Interior-point methods (IPM). However, the computational complexity of IPM applied to SOCP with *dense* matrices grows rapidly with problem dimension, so that large problems of this type arising in signal and image processing are well beyond the reach of these techniques. On the other hand, these problems possess nice geometry associated with complex $\ell_1$-norm. Moreover, their *first-order information* – the value of objective and its gradient at a given $\varphi$ – can be computed using Fast Fourier Transform in time which is almost linear in problem size. Therefore, we used first-order optimization algorithms, such as Mirror-Prox and Nesterov's accelerated gradient algorithms (see [14] and references therein) in our recovery implementation. A complete description of the application of these optimization algorithms to our problem is beyond the scope of the paper; we shall present it elsewhere.

**Interpolating recovery**　In Sec. 2-3 we considered only recoveries which estimated the value $x_t$ of the signal via the observations at $n + 1$ points $t - n, ..., t$ "on the left" (filtering problem). To recover the whole signal, one may consider a more flexible alternative – *interpolating* recovery – which estimates $x_t$ using observations on the left *and* on the right of $t$. In particular, if the objective is to recover a signal on the interval $\{-n, ..., n\}$, one can apply interpolating recoveries which use the same observations $y_{-n}, ..., y_n$ to estimate $x_\tau$ at any $\tau \in \{-n, ..., n\}$, by altering the relative position of the filter and the current point.

**Blockwise recovery**　Ideally, when using pointwise recovery, a specific filter is constructed for each time instant $t$. This may pose a tremendous amount of computation, for instance, when recovering a high-resolution image. Alternatively, one may split the signal into blocks, and process the points of each block using the same filter (cf. e.g. Theorem 2.1). For instance, a one-dimensional signal can be divided into blocks of length, say, $2m + 1$, and to recover $x \in \mathbb{C}(\mathbb{Z}_{-m}^m)$ in each block one may fit one filter of length $m + 1$ recovering the right "half-block" $x_0^m$ and another filter recovering the left "half-block" $x_{-m}^{-1}$.

# 7　Conclusion

We introduced a new family of estimators for structure-blind signal recovery that can be computed using convex optimization. The proposed estimators enjoy oracle inequalities for the $\ell_2$-risk and for the pointwise risk. Extensive theoretical discussions and numerical experiments will be presented in the follow-up journal paper.

## Acknowledgments

We would like to thank Arnak Dalalyan and Gabriel Peyré for fruitful discussions. DO, AJ, ZH were supported by the LabEx PERSYVAL-Lab (ANR-11-LABX-0025) and the project Titan (CNRS-Mastodons). ZH was also supported by the project Macaron (ANR-14-CE23-0003-01), the MSR-Inria joint centre, and the program "Learning in Machines and Brains" (CIFAR). Research of AN was supported by NSF grants CMMI-1262063, CCF-1523768.

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
