[Reviews · NeurIPS 2016]

Reviewer 1

Summary

The paper studies theoretically linear filters for signal denoising. An interesting formulation of the problem is obtained under the assumption that the signal has a regular component in an unknown shift-invariant linear subspace, a bounded free component and an i.i.d. Gaussian noise component. It is proposed that the recovery formulation that constrains or penalizes l1 norm in the Fourier space, under some technical conditions, possess guaranteed performance bounds. The goal here is to obtain an estimator achieving a risk within a constant additive bound (called the price of adaptation) of that of the optimal linear estimator. The paper improves upon previous works [10,5] providing tighter additive bounds. Penalized recovery version does not need to know the filter norm parameter in advance.

Qualitative Assessment

I’ve indicated below that I am not confident in this topic, please take my questions with ease. Clarity A general question about blind filtering: Is it indeed the case that the knowledge about the set of signals X cannot be obtained / learned in some applications or it is more that it is difficult to describe? In the case of shift-invariant subspaces the description does not seem to be a problem, it is given by equations (6). Small-error linear filters. It was unclear why the signal is restricted to the support interval [0 n] but its recovery is requested on the interval [-m 2m]. Does it mean in particular that there are m samples of pure noise available? How does this support assumption combine with shift-invariance equations (6)? The simplicity assumption is designed such that the total recovery error (5) does not depend on the length of the sequence, and the filter size must grow proportionally to the sequence support. It does not seem realistic. While the main theoretical contributions were presented for the “sequential” filters – only signals to the left on the axis are participating, the experimental results are presented in the different setting – full 2D linear filters. Is this difference not essential? Do the bounds continue to hold? There are many image denoising techniques in the literature. If we compare to TV-denoising, the prior is imposed on the reconstructed signal, whereas in the proposed method it is imposed on the filter. Can the methods be theoretically / experimentally compared? Since the reconstruction requires to solve an optimization problem, is the special structure of requesting a linear filter is still well-justified / beneficial in practice? While the discussion section dives deeper in a specific direction, a concluding section discussing the theoretical results and experiments overall is missing. Technical Line 56: a new notation seems to be introduced, in which case “may be written as” is not appropriate. Line 75: \phi^0 Line 93: \Delta^\tau is not defined – compositional power? *************************** I would recommend to the authors to improve on accessibility of their work. While presented results are really theoretically interesting, understanding them is not easy. In particular the assumptions made for the theory to hold are by far non-trivial. Try to show exemplar cases where these assumptions do hold and if possible explain more their meaning. Do these assumptions translate to something that can be calculated post-optimization so that a provable bound for a quality of reconstruction of a given instance can be computed? It would be also beneficial if the handling of different kinds of filters w.r.t. time axis was more uniform across the paper. Despite the authors claim the results are easily extendible, it is not straightforward to see how the assumptions and the bounds should extend to 2D full filters. The question regarding optimization-based recovery of linear filter versus optimization-based recovery of the signal directly was not fully answered in the rebuttal. In particular problem (8) can be rewritten with a substitute variable \hat x = \varphi * y, which will result in a special regularized on \hat x. While the method does not need to know the structure (the shift-invariant linear subspace) of the signal, in many applications such structure, e.g. appropriate basis can be learned from the training data. The advantage of doing blind recovery in this setting is not of that high practical interest in my opinion.

Confidence in this Review

2-Confident (read it all; understood it all reasonably well)


Reviewer 2

Summary

The paper proposes a new estimator to recover signals from Gaussian noise, which works on the Fourier representation of the signal. The benefit is to eliminate some previously made assumptions, but still assumes that a linear model captures the signal, and improved risk bounds. Some preliminary numerical experiments are presented.

Qualitative Assessment

Figure 4 is too small and is hard to read. A statistical analysis is missing, the authors are right in saying that these results are preliminary. I was wondering how would the results compare with a simple Wiener filter? I missed a Conclusions section.

Confidence in this Review

1-Less confident (might not have understood significant parts)


Reviewer 3

Summary

In this paper, the authors investigate the L2-risk of some linear filters when aiming at filtering a noisy finite sequence of observations. They propose two recovery methods (least-squares constrained or penalized by L1 norm in the Fourier domain) with a tuning parameter. They prove exact oracle inequalities for the L2 risk comparing the risk of their estimator to the risk of a linear filter $\varphi_0$ with small L1 norm in the Fourier domain. Their theoretical analysis asserts the L2-risk in terms of the hidden structure (Assumption A) and the L2-risk of the best linear filter with prescribed L1 norm in the Fourier domain ($\varphi_0$).

Qualitative Assessment

In Figure 2 (and in Figure 1 in a less obvious manner), one can see that four squares appear in the picture of the penalized recovery method. This "artefact" is somehow disturbing. To my point of view, it might be due to the fact that the authors use a filtering estimator. Is it possible to extend your analysis to "smoothing"/"interpolation" estimators that recover $x_t$ from the full sequence $y_{-n},...,y_n$?

Confidence in this Review

2-Confident (read it all; understood it all reasonably well)


Reviewer 4

Summary

The paper is about the estimation of a signal under additive gaussian noise. The authors generically assume that linear estimators work well without assuming any knowledge of the underlying signal. They aim to propose estimators that perform as well as the best linear estimator.

Qualitative Assessment

I felt that the paper can be written more clearly. I had some trouble understanding the main points. For example, the setup introduced at the beginning of the paper is that of estimating a signal x_0, ..., x_n in sequence from y_{-n}, ..., y_{-1}, y_0, .., y_n (while estimating x_t, one can only use the data y_{-n}, ..., y_t). It appeared to me that, in this setting, one cannot use y_{t+1}, .., y_n to estimate x_t. But the estimator used by the authors employs the entire data to estimate x_0, ..., x_n (because \hat{\phi} depends on all the y_ts). This sequential estimation setting is also not applicable in the authors' examples where time seems to be two-dimensional). The main theorems are proved under Assumption A and the assumption that x is (m, \rho) simple. I did not find both these assumptions intuitive. Also it is unclear why these need to be satisfied a priori (this needs to be clarified because the authors claim to be doing a blind recovery of the signal).

Confidence in this Review

2-Confident (read it all; understood it all reasonably well)


Reviewer 5

Summary

The paper proposed a family of estimators to recover signals with Gaussian noise. Error bounds are deduced for both constrained recovery method and penalized recovery. Several numerical experiments are used to show the effectiveness of the method.

Qualitative Assessment

Although the authors provide an example for the Assumption A, it does not seem to be a natural or verifiable condition. The main results are largely impacted by those in [5, 9, 10]. The only notable improvement on the price for adaption. The equivalence between the constrained recovery and penalized recover is somewhat well known in the field of inverse problem and regularization theory. The experiments are insufficient. As the main methods and results are closely related to the reference [5, 9, 10]. Comparison with algorithms in these references are not well performed. The notation phi(Delta) should be explained in line 56. It is better to emphasize epsilon in the Assumption A is part of the signal to be recovered. Actually I would suggest a different notation so that it does not confuse readers to regard it as certain noise.

Confidence in this Review

2-Confident (read it all; understood it all reasonably well)


Reviewer 6

Summary

The paper provides an extension of the works in [9] and [5] cited in the paper to the \ell_2 risk case. The results are interesting and informative. I believe the experimental evaluation is a bit lacking since performance comparisons are only performed for atomic norm soft thresholding in most experiments. Since the work is primarily an extension of [9,5], I believe that the main performance comparison should be with respect to the methods of [9] and [5].

Qualitative Assessment

The extension of the work in [9] and [5] to the \ell_2 risk scenario is important for denoising problems. As mentioned earlier, I believe the experimental evaluation is a bit lacking since performance comparisons are only performed for atomic norm soft thresholding in most experiments. Since the work is primarily an extension of [9,5], I believe that the main performance comparison should be with respect to the methods of [9] and [5]. Concerning the paper presentation, my impression is that too much information has been packed into the 9 page paper. Perhaps the paper is more suitable as a JMLR manuscript where details can be elaborated. That said, I still think that the paper presents an important enough contribution to warrant publications in NIPS. I also have a couple of editorial comments: - In equation (3), please specify the variable over which the expectation is computed. - Equation before line 93: the x looking character is not defined. - In equation (7), please add a sentence clarifying that \hat{x}_{con} is as defined in (2).

Confidence in this Review

2-Confident (read it all; understood it all reasonably well)


Reviewer 7

Summary

This paper suggest a signal restoration technique using linear estimators and provides theoretical guarantees on the performance under some mild assumptions. The proposed practical performance are quite weak. However, this is due to the fact that the recovery is performed with a linear estimator, which is very limited from the beginning.

Qualitative Assessment

The analysis performed in this paper is only for liner estimators. As these estimators are very limited and most of the recovery techniques used today are non-linear, a part that would explain the implication of the proposed results also for non-linear estimators is missing. Indeed, the main interesting part in the paper is the theory developed for the linear estimators. However, as these estimators are very limited in their usage, a part that would show how the developed results can be more broadly used is missing in the paper.

Confidence in this Review

3-Expert (read the paper in detail, know the area, quite certain of my opinion)